# Cohesion: A Measure of Organisation and Epistemic Uncertainty of Incoherent Ensembles

**DOI:** 10.3390/e25121605

**Published:** 2023-11-30

**Authors:** Timothy Davey

**Affiliations:** Research Department, London Interdisciplinary School, London E1 1EW, UK; tim.davey@lis.ac.uk

**Keywords:** uncertainty, complexity, aleatoric, epistemic, systems, replicability, ergodicity, indeterminacy

## Abstract

This paper offers a measure of how organised a system is, as defined by self-consistency. Complex dynamics such as tipping points and feedback loops can cause systems with identical initial parameters to vary greatly by their final state. These systems can be called non-ergodic or incoherent. This lack of consistency (or replicability) of a system can be seen to drive an additional form of uncertainty, beyond the variance that is typically considered. However, certain self-organising systems can be shown to have some self-consistency around these tipping points, when compared with systems that find no consistent final states. Here, we propose a measure of this self-consistency that is used to quantify our confidence in the outcomes of agent-based models, simulations or experiments of dynamical systems, which may or may not contain multiple attractors.

## 1. Introduction

Most experiments have two types of uncertainty [1]. The first is *epistemic*, which can be *reduced* through experimentation. For instance, if you are given a coin, but do not know whether it is biased, you can gain confidence about the nature of its bias through repeated trials. Specifically, if it were measured that out of 1000 coin tosses 80% were heads, the epistemic uncertainty would be reduced by concluding the coin was biased toward heads. The second, *aleatoric* uncertainty, is *irreducible* through experimentation. In the case above, even if you know there is a bias toward heads, the coin will still land on tails ≈20% of the time, thereby always leaving some fundamental aleatoric uncertainty about the outcome of the next toss.

*Incoherence* [2] introduces a new type of uncertainty. There is often an assumption underlying the scientific method that, for a known and consistent set of macro-parameters, the outcome of an experiment under lab conditions will always be the same. Unfortunately, complex dynamics [3] often drive systems to be *non-ergodic* [4], thereby breaking this assumption. This means that running the same experiment or simulation multiple times (samples), with identical macro-parameters, can lead to dramatically different results due to compounding non-linear interactions within the system. This is sometimes called Indeterminacy.

This is visualised in Figure 1. Set A is a illustrative distribution of values with low aleatoric uncertainty as the points are clustered together, but high epistemic uncertainty because there are only a few measurements (where both of these are measured relatively). You can decrease the epistemic uncertainty by gathering more data, which is Set B. This extra data might decrease the aleatoric uncertainty (which can be thought of analogous to variance) if the points are tightly bound (again like B) or it might increase if not (like C).

However, there is another type of aleatoric uncertainty, called Incoherence [2], which is illustrated by Set D. Where a system might produce low variance distributions each time it is run, but the difference between those distributions is large. For instance with a climate model with complex dynamics and tipping points. Therefore, this uncertainty is generated not by the distribution itself but instead by *which* distribution the system will manifest.

Having said that, through simulation, it is possible can gain a better understanding of the distribution of distributions (e.g. the placement of D sets), thus making that uncertainty partially epistemic. This is why there will always be uncertainty arising from climate models, but there is a lot of value in estimating the landscape of possible end-states.

The aim of this paper is to introduce a measure of this specific epistemic uncertainty. Since it is fundamentally driven by how grouped the distributions are into distinct scenarios, we call this *Cohesion*.

Incoherence is defined [2] by looking at the spread of entropies ($H_{k}$ for each sample distribution *k*) around the entropy of the point-wise mean distribution $\tilde{H}$ (where the results across all samples are pooled together).
(1)$$I=\sqrt{\frac{1}{\tilde{H}}\sum_{k}w_{k}{\left(\tilde{H}-H_{k}\right)}^{2}}$$

## 2. Idealised Use Case

To demonstrate why cohesion is useful, we shall use an idealised use case, in which you own a silicon chip manufacturing plant and want to upgrade the system. Using a digital twin, you test two options. Option A involves a novel queuing system, and Option B involves combining two machine types into a single new machine. These options are visualised in Figure 2.

Both options show similarly high levels of incoherence. A low incoherence implies a high *prior* predictability, that is, before you run or inspect the system, you have a high confidence in which eventual state or equilibrium the system will take. A low incoherence might be important, for instance, in aircraft or space shuttle launches, where the state in which that flight will result impacts the level of fuel and other supplies that need to be considered before launch. A low incoherence, in our case, might be important because of being able to manage lean raw material supply chains. This means that, counter-intuitively, staying with the current option, which has a low incoherence, might be the best option.

Cohesion offers a level of *posterior* predictability, that is, once the system is running, the likelihood of having *self-organised* into a specific scenario. For instance, systems with tipping points and attraction basins will have a high cohesion.

In this example, looking at Option A, the mean μ of the pooled distribution and the best case distribution offer significant improvements to the current setup. However, since it has a low cohesion, the exact performance you are likely to obtain for each instance of running the setup is very unknown.

In contrast, in Option B, the possible distributions fall into two very clear scenarios, leading to a high cohesion. On the face of it, it has a high incoherence *I* and, therefore, might put you off, similar to flight or space launches, especially as the pooled μ offers no improvement. However, its high cohesion might compensate for its high incoherence.

First, if you are able to reset your system with minimal wastage or downtime, then it is much more likely that the system will evolve into a well-known scenario, enabling you to quickly find the optimum.

Second, if restarting the system is not possible, as with climate change models or flight systems, then, in cohesion cases, you at least have a defined and specific set of scenarios to build contingencies around. This means that the chance of obtaining the optimum scenario might be worthwhile if there is a cost-effective mitigation that you can put in place to compensate if you obtain an unwanted scenario.

Third, in high-cohesion scenarios, leading indicators are much more likely to exist. For instance, in Option B, since these distributions do not overlap, early measurements will give a clear guide as to which scenario the system is in. This means that if you are able to respond to leading indicators quickly, then high-cohesion and high-incoherence scenarios might also still be preferable.

## 3. Philosophical Interpretation

Beyond aleatoric and epistemic uncertainties, these measures can be thought of in terms of the frequentist and Bayesian interpretations of statistics.

The frequentist interpretation of the probability of rolling a die is strictly asking “out of all the infinite possible parallel futures, for this next throw what proportion will land side *X*”. Unfortunately, we cannot access all of these parallel futures once an event has occurred. Therefore, to make this interpretation practical, we approximate it by using the many historic measurements made in serial.

However, this can be troublesome. For instance, take a coin which on its first throw has a fair probability of landing on heads or tails. However, all subsequent throws slightly favour the previous side; it then resets after a few days of inactivity. In our experiment, if the first throw happened to be heads, we might incorrectly conclude the coin is biased toward heads, rather than whatever the previous throw was. Since incoherence is a measure by which a single (serial) distribution accurately represents all possible (parallel) distributions, you could see incoherence as a measure of how effective the frequentist interpretation is in a particular case.

Similarly, cohesion measures the information gained about a system, given information about that system. That is, in systems with high cohesion, information about the distribution of the current state of the system gives more information than in low-cohesion systems. Therefore, you could say that cohesion is a measure of how effective the Bayesian interpretation is for that case.

## 4. Cohesion Formulation

Cohesion is “the act or state of sticking together tightly” of the distributions into scenarios, in this case. Therefore, the first step is to determine the notional distance between each sample with all of the others (in this context, it is important to understand there is no expectation or central point from which to measure). We do this by calculating the Jensen–Shannon divergence [5] *d* for all dual pair-wise a,b combinations of samples, creating the set of divergences *D*:(2)$$d_{ab}=w_{a}\left({\tilde{H}}_{ab}-H_{a}\right)+w_{b}\left({\tilde{H}}_{ab}-H_{b}\right)$$
where $H_{a}$ is the *information entropy* [6] of sample *a*. ${\tilde{H}}_{ab}$ [2] is the entropy of the pooled sample, where observations from *a* and *b* are combined to make a new single distribution (the point-wise mean). *H* (the entropy estimator) is defined on a system by system bases based on which is most appropriate for the data, i.e., various entropy estimators are available for continuous distributions beyond Shannon’s original definition (which is appropriate only for discrete data).

It should be noted the Jensen–Shannon divergence was specifically chosen over the Kullback–Leibler divergence. This is because, when you have two distributions that have zero values in common (which might be possible, for say, temperatures around a tipping point), the latter results in zero or infinite values, making this version of the metric inconsistent at best and unreliable at worst.

We then weight the *w* of these such that $\sum_{i}w_{i}=1$, which was chosen based on what is known about the data. For instance, w=1/2 is appropriate when you know there is a bias in the sample volumes between samples, but there are still sufficient data to fully represent each sample. Alternatively, when you are re-sampling from a single distribution, using the count of observations for that sample, *n*, over the observations for both samples, *N*, means w=n/N is likely a more appropriate choice.

To estimate how clustered the outcomes are, we use the information entropy [6] of the calculated divergences. This is because we are interested in the *epistemic information gain* from how much the outcome distributions cohere pair-wise. Entropy is a direct reflection of this, particularly as it takes into account the density of divergences and not just the spread.

To calculate the entropy of this continuous set of divergences, we use the entropy estimator “Density Variance”. See Figure 3 and Figure 4. This technique involves calculating the Kernal Density Estimates *G* for *R* total evenly distributed individual reference points *r* such that, given *N* total individual observations *o*, the reference point *j* has a KDE $g_{j}$ of
(3)$$g_{j}=\frac{1}{N}\sum_{i}^{N}e^{-k∥r_{j}-o_{i}∥}$$
Due to the concave nature of the entropy curve, these densities overemphasise the difference between small divergences. Therefore, to linearise this measure, we take the square root of $g_{j}$, where appropriate practical constants are found to be k=100 in Equation (Equation 3), with R=500 reference points evenly distributed over [−0.5,1.5] when the divergences *D* are normalised. This leads to a continuous entropy estimator of
(4)$$H=1-\frac{\hat{Var}\left(\sqrt{G}\right)-V_{0}}{V_{1}-V_{0}}$$
where $V_{0}$ is the variance of the density for a maximal entropy distribution (i.e., uniform distribution) and $V_{1}$ is the same for a minimal density distribution (i.e., all observations are 0). Using the constants above, it is found that $V_{0}=0.0047683913076489465$ and $V_{1}=0.009711690148675985$ by allowing H to be bounded by [0,1], where values outside this can be minimally and maximally bounded.
(5)$$C={\left(1-H\left(\sqrt{G}\right)\right)}^{2}$$This ensures that the cohesion is maximally C=1 when all distributions are identical and C=0 when there is high entropy, i.e., a high variance in divergences between samples and, therefore, limited consistency between sample distributions.

**Figure 3 entropy-25-01605-f003:**
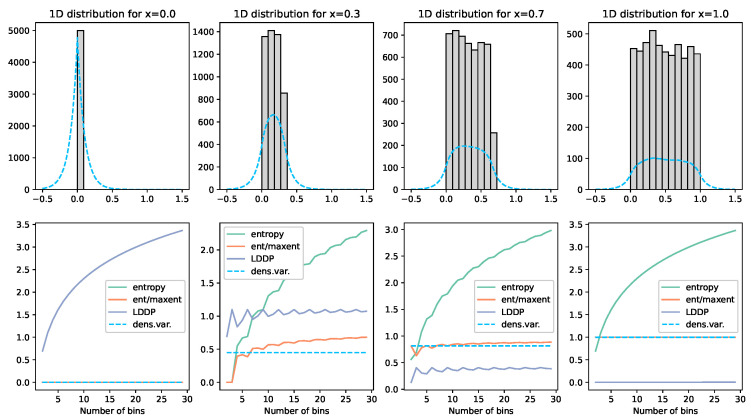
Illustration of the density variance technique and comparisons. Here, we demonstrate on a uniform distribution of 5000 values [0,x]. The grey histogram visually illustrates the placement of these values. The blue dashed line displays the Kernal Density Estimator used to approximate the shape of the distribution. Each graph column represents a different *x* value. The graphs below each then show the various measures calculated. The green line labelled entropy calculates the entropy of the points for the given number of bins on the x-axis over the range [0,1]. The orange line normalises this value by the maximum entropy for that given number of bins. The grey LDDP line is Jaynes’ Limiting Density of the Discrete Points method. The blue dashed line is, the density variance method described above. What we can see here is that the LDDP is firstly not normalised; secondly, it is lowest when the entropy should be highest; and, finally, it is not bin-independent. The normalised entropy (orange) gives a reasonably consistent value for entropy, but, with low bin numbers (which would be required with low observation numbers), this value becomes very unstable. The density variance (dashed) tracks this orange line remarkably well and is entirely bin-independent. The green line on the bottom left graph is obscured under the orange line at zero.

For the exact details of how this equation has been implemented, the Python code for these results has been made available open source (https://github.com/timjdavey/incoherence, accessed on 30 October 2023).

## 5. Method

### 5.1. Organised Noise

Here, we are looking at the cohesion’s ability to explicitly identify organisation. Here, we model a detection system that receives a series of different frequency bursts over time. The team wants to understand how likely the bursts are to be an organised pattern from an unknown source. The language or cypher is not known, so they want to avoid longitudinal analysis until they have more certainty. None of the bursts are identical, as there is noise in the system caused by transmission or creation. We can think about each burst as a letter in an alphabet, and we are trying to discern whether there are specific letters being transmitted or whether this system is simply noise.

Each letter is composed of 2 Gaussian distributions. The properties of these distributions were randomly calculated; the mean was uniform [0,20), the standard deviation was a random normal pick around 1 of 0.5, and the sample count was a random int [1,10). For each timestep, we then randomly pick from the alphabet of letters and, again, randomly generate the corresponding data points based on that letter’s parameters.

The only deviation from this is the manual adjustment to the top variant graph so that we could compare similar but not identical datasets. Overall, this system generates a message using a set of letters that contains a reasonable amount of noise.

### 5.2. Daisyworld

The Daisyworld model was proposed by James Lovelock to investigate the *Gaia hypothesis* [7] in which a toy flat world and sun are created such that it demonstrates that, if the world is populated with species of daises with varying albedos (how much they reflect light from the sun), they will undergo a complex set of dynamics such that they will *self-organise* to stabilise the temperature (in the time dimension) of the planet to a level that is most hospitable for those daisies.

The model used here is ported from the Netlogo version [8] to Python and is open-source (https://github.com/timjdavey/incoherence).

In our case, the world is populated by two species of daisies. The black and white daises had albedo α values of 0.25 and 0.75, respectively (the standard level of this model), while empty spaces had an albedo of α=0.4. Higher albedos α reflect light away from the planet, cooling the local area, and darker regions absorb more light, raising the local temperatures. The daisies can only survive within the local temperature range of [5,40] degrees. If they do survive a given timestep, each daisy will populate randomly into neighbouring cells or, after a given (old) age, they will die.

The main parameter of the model is the luminosity *l* of the sun, as this has the greatest impact on how the daisies self-organise to create an optimal global temperature. For illustrative purposes, we sampled this luminosity over the range 0.45≤l≥1.45. For each value, we conducted 30 identical simulations, running each for 400 timesteps, which is enough for the system to comfortably reach equilibrium. Each sample started with a random dispersion of 30% white, 30% black, and 40% empty points on a 29×29 grid world.

## 6. Results

### 6.1. Organised Noise

Looking at Figure 5, the top graphs are the most organised with only two letters and have a high cohesion value *C* of 0.80. Figure 6 can be used to identify the underlying letters behind each burst, represented by their colour. The topmost graphs are highly organised but would only achieve a cohesion value of 1.0 if the two patterns were identical. However, there is no prescription for the distribution of those points. For instance, at first glance, the top left graph looks like the patterns are identical; however, the proportion of points at the higher value vs. the latter is different, as well as the spread, meaning the probability density functions and entropies of the distributions are different. However, as Figure 2 further illustrates, even two single-mode scenarios create a cohesion value of 0.8.

The bottom left graph has some organisation, but this time, it is organised into five patterns and, therefore, has a lower cohesion value to represent this. In contrast, on the bottom right, there are no set pattern, leaving the lowest cohesion value.

In the Daisyworld model, we use the standard deviation σ of the discrete data, the standard deviation of the means σ(μ) and the standard deviations σ(σ) of the samples as measures of the lack of organisation in a set of samples. As we shall see in such a case, these measures reflect that organisation well, despite the downsides of poor interpretability and a lack of consistency across discrete and continuous data. However, in this case, it can be seen that these measures are a poor indicator of organisation. Where none of these measures correlate with Cohesion or the underlying organisation of the patterns. This is because those measures are only truly effective if you consider a single mode as your definition of organisation.

### 6.2. Daisyworld

Here, we look at the cohesion measure in the context of our example Daisyworld toy model.

Looking at the distribution of temperatures shown in Figure 7, we can observe the following key takeaways:The balancing feedback loops within the system do cause the temperatures to organise within the temperature band (represented by the grey dotted lines), which is optimal for daisy survival.However, given this, and given that the model is built deliberately to demonstrate self-organisation, the outcomes vary significantly for some luminosity values as seen by how much the mean global temperature (blue dots) of each individual simulation can vary. This demonstrates that incoherence [2] exists for this model.In particular, both incoherence and cohesion vary depending on the luminosity value.Despite this incoherence and the lack of consistency among simulations (samples), the mean of the means (red line) and the mean of the pooled observations (orange) are remarkably similar for the entire luminosity range, even in the notably low cohesion range where the luminosity *l* is 1.2<l<1.3. This demonstrates how cautious we must be in comparing distribution means to determine homogeneity.We see that the incoherence (purple) finds three key tipping points in the model.The leftmost tipping point is where the system goes from no plants surviving to some black daisies surviving (see Figure 8). It should be noted there is not a clean cut-off in the parameter causing this tipping point. However, instead, like most real-world tipping points, they exist for a small *range* of luminosity values (0.50<l<0.55) where the system attracts into two stable outcome states with clearly identifiable mean temperatures. Since these states are so different, the incoherence between them is high. However, since these states cohere into just two clean scenarios, we can see the cohesion is still relatively high (shown by the low teal 1-cohesion line).The rightmost tipping point (1.1<l<1.3) is where the black daisies begin to no longer survive. Here, we see this tipping point (much like the left) involves, at first, two clear, distinct states. However, unlike the left, this tipping point spans a much greater parameter range where, in particular, at higher luminosity values, we see a breakdown of self-organisation with no clear set of scenarios into which the system stabilises. This results in the cohesion dropping dramatically (i.e., the 1-cohesion teal line spikes).The central region where both species survive (0.9<l<1.1) shows a markedly higher spread of mean temperatures (blue dots) compared to the region where only black daises survive (0.55<l<0.75). Despite this, the incoherence is still low because there is still a high overlap between the underlying distributions.We also see the value of cohesion in this context. Here, we see the leftmost tipping point falls into two distinct scenarios, causing the incoherence to be high, as these distributions are very different. It also causes the 1-cohesion to spike, though not by much, since the scenarios are very clear.Meanwhile, the rightmost peak does not have any clear scenarios with a wide spread of blue dots and higher standard deviation of standard deviations. This shows that this critical point is of a vastly different nature.

**Figure 7 entropy-25-01605-f007:**
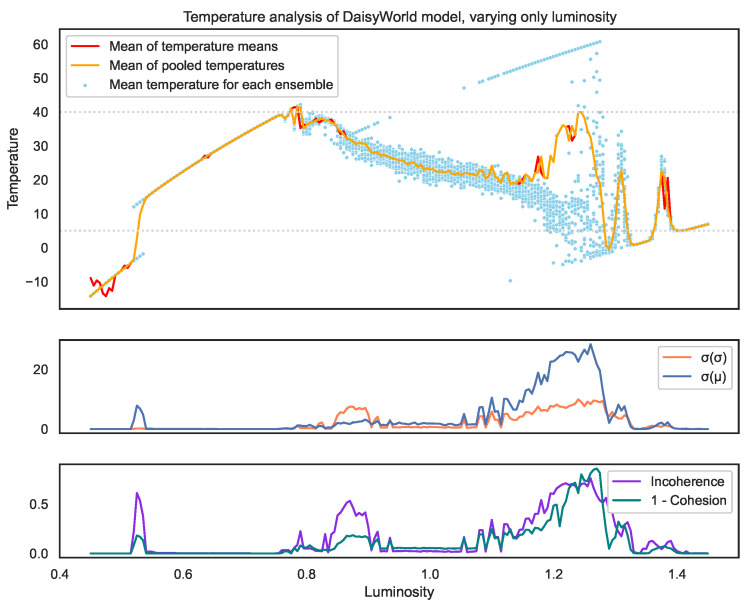
The top graph shows the final temperature values (after 400 timesteps), for a range of the key model parameter (sun luminosity). The model outputs a distribution of temperatures for across the globe; the blue dots represent the mean of these temperatures for each of the 30 samples run for each luminosity value. The red line is the mean of these blue dots. The orange line is the mean of all the temperatures of the single (pooled) distribution. The second graph shows the standard deviation of these temperature means (blue dots) σ(μ) and the standard deviation of the distribution of temperatures σ(σ). The third graph shows the incoherence and incohesion (1-cohesion) values. Although, in this case, both σ(σ) and σ(μ) are somewhat correlated with cohesion, the shapes of their graphs show them as poor measures. First, it is worth noting that incoherence and incohesion are correlated because a system must be incoherent for it to be incohesive by definition. Cohesion measures how much organisation there is between those different distributions, whereas incoherence measures how different those distributions are. The value 0.53 (left grey dotted line) shows that σ(σ) is a poor measure of cohesion since it detects that the distributions are similar in shape but fails to detect the difference in their means. σ(μ) naturally detects this difference but then fails to demonstrate a low consistency at the middle grey line. Meanwhile, 1.28 (the rightmost grey dotted line) shows the best illustration of cohesion, where we see the enormous spread in means (blue dots) compared to 0.53, where they are clustered into 2 clear scenarios.

Looking then at the corresponding distribution of species counts shown in Figure 8, we now observe the following further takeaways:The incoherence and cohesion values using only the distributions of species counts are remarkably predictive of those for the temperature distributions. In particular, they find the same three tipping points with the same characteristics.While the standard deviations (second graph) of the counts do align somewhat with the tipping points, they do not align with the standard deviation of the temperature means. Moreover, since they represent a multivariate distribution, it is harder to interpret what this means in terms of the overall model certainty.

**Figure 8 entropy-25-01605-f008:**
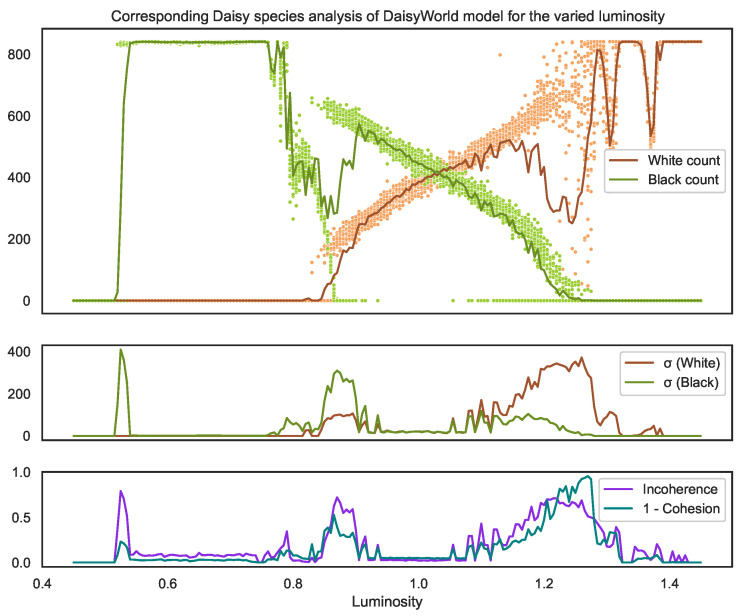
The corresponding species plot for Figure 7. The top graph displays the count of black daisies (green dots) and white daisies (brown dots) for each sample, where the line of the respective colour on the top graph represents the mean of these counts and the line on the second graph represents the standard deviation. It is remarkable that despite using different subset distributions and types of data (continuous vs. discrete), both the incoherence and cohesion measures of the whole system are so similar.

## 7. Discussion

Here, we have outlined *cohesion*, which is a measure associated with the amount by which you can potentially reduce epistemic uncertainty arising from a sample *incoherence*. It is an important measure by which to quantify and interpret the results from simulations. It offers a clear vocabulary for policy- and decision-makers to understand and articulate the certainty of given scenarios, even if those scenarios are very different.

It has previously been shown [9] that nearly all dynamical systems that have an attractor can be described as self-organizing [10] if we choose to label that attractor as “organized” [11]. However, if organisation were to only occur occasionally, we would label it a “fluke” (e.g., seeing a portrait of Jesus in toast), or, if it occurred predictably and mechanistically in parameter space, we would label it “deterministic” (e.g., the boiling point of water at varying pressures). Therefore, for us to label it as self-organisation, the system needs to self-organise *repeatably* into similar (but not identical) structures, under varying conditions. Finally, these repeated structures must be low-entropy since, due to thermodynamics, high-entropy organisation (e.g., gas uniformly distributing itself over an entire volume) is both the dominant state of any system and the dominant *state-path* away from any other low-entropy organising state that we might impose, which is therefore typically (and inappropriately) labelled as “disordered”. Therefore, it needs to be a low-entropy state; otherwise, the organised (and often diffuse state) is labelled as disordered.

Therefore, it can be argued that cohesion is a measure of self-organisation, as it quantifies the existence of system attractors that stabilise samples into repeatably identifiable low-entropy scenarios.

This leads *incoherence* and *cohesion* as a metric couplet to be extremely useful across a variety of use cases to identify and quantify types of uncertainty that are becoming extremely common thanks to the availability of large computations and simulations.

## Figures and Tables

**Figure 1 entropy-25-01605-f001:**
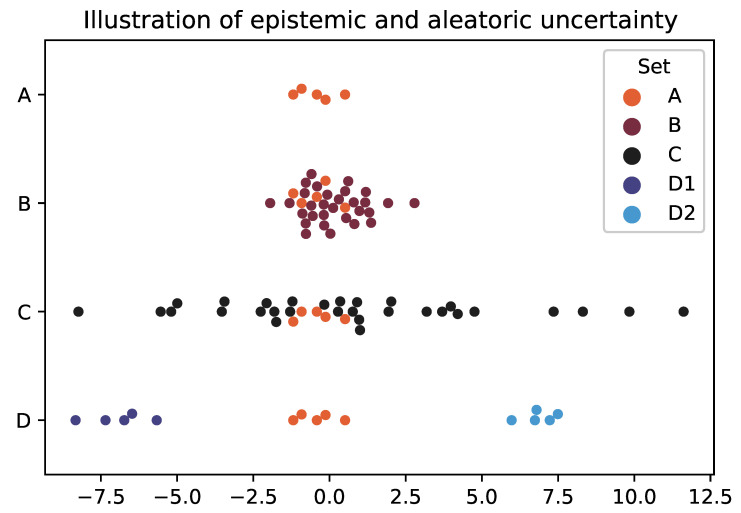
Category A has a low aleatoric uncertainty as the points are clustered together and a high epistemic uncertainty as there are so few measurements. B and C both have lower epistemic uncertainty as they both have many more points. However, B has a lower aleatoric uncertainty than C because it has a lower variance. D is an example of incoherence, where although each distribution might have a low variance, since you do not know which distribution your system will result in, the overall variance is high, leading to a high aleatoric uncertainty. This is a swarm plot, where any variance in the y-axis within the category is for visibility purposes.

**Figure 2 entropy-25-01605-f002:**
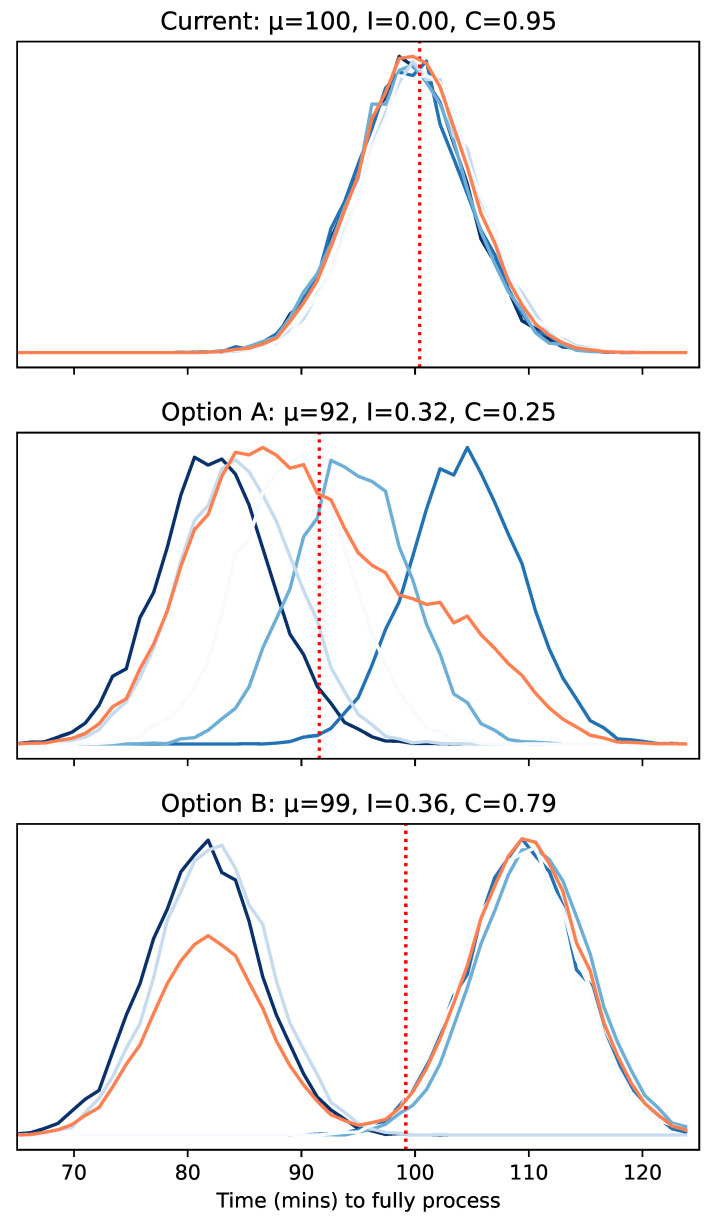
Illustrative results for visualisation purposes in the introduction. These are meant to represent hypothetical results from a digital twin of the manufacturing plant making silicon chips as described in the Idealised use case. The distributions represent the steady-state times that it takes for chips to be produced from end to end. Each graph shows the individual probability density functions of five trial simulations in blue. The pooled distribution is shown in orange, with its mean drawn as the dashed red line. The title displays the value of the pooled mean μ, the incoherence *I* and the cohesion *C*.

**Figure 4 entropy-25-01605-f004:**
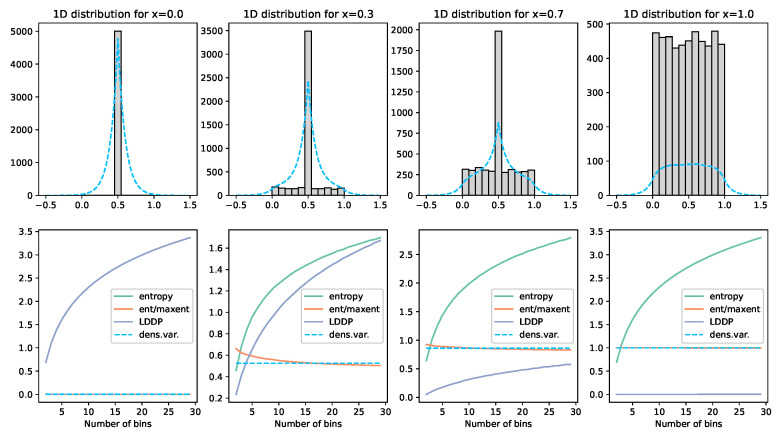
See the description in Figure 3 for details. The distribution displayed is 5000(1−x) points at 0.5 and 5000x points uniformly distributed between [0,1]. Here, we see again that the density variance (dashed) line is invariant to the bin count and approximates the normalised entropy value (orange) from a far lower bin count. The green line on the bottom left graph is obscured under the orange line at zero.

**Figure 5 entropy-25-01605-f005:**
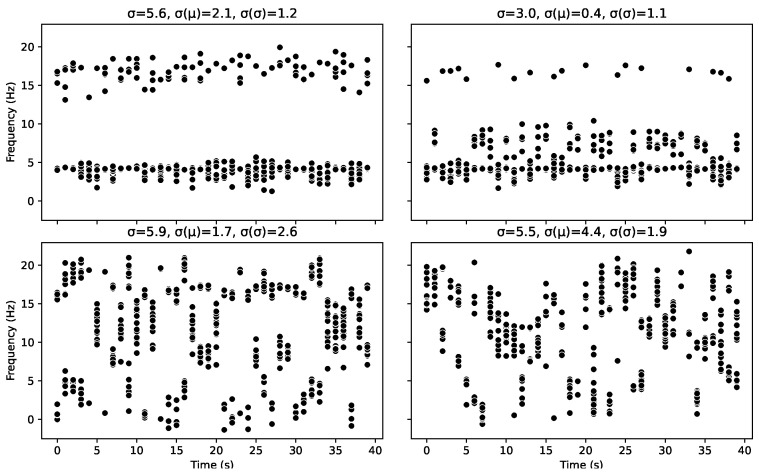
Four messages composed of different numbers of generating burst patterns (which can be thought of as letters or symbols in a message). The top left shows a binary (two-pattern) message; the top right uses a very similar two-pattern message. The bottom left has five letters, while the bottom right has a unique burst pattern for each timestep (i.e., no patterns). This graph also outputs the values of the standard deviation of the entire dataset σ, the standard deviation of the means of each pattern σ(μ), and the standard deviation of the standard deviations σ(σ). Here, we see that none of these measures correlate to the organisation of the message in terms of number of the patterns or the letters that it contains.

**Figure 6 entropy-25-01605-f006:**
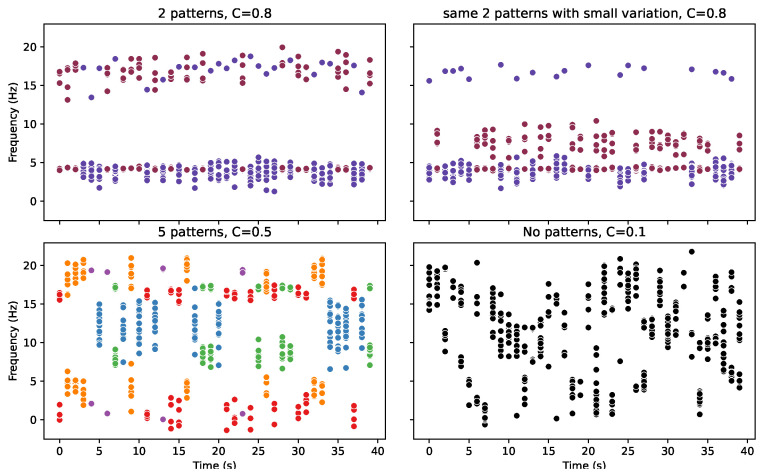
Figure 5 is now coloured by the burst pattern. Where each colour represents a different underlying pattern. Where the Cohesion value C is given in the title. These graphs firstly demonstrate clearly that not only is cohesion effective at quantifying the organising nature of the patterns but also that it is extremely easy to interpret the measure (going from 1 highly organised to 0 not organised). What the top two graphs are designed to demonstrate is the nature of the organisation. The cohesion comes from the fact there are only two patterns, while how similar those patterns are is barely considered.

## Data Availability

All data generated and code used and/or analysed during the current study are available in the incoherence repository at https://github.com/timjdavey/incoherence.

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
