# Peer review of "Cohesion: A Measure of Organisation and Epistemic Uncertainty of Incoherent Ensembles"

_entropy, 2023, doi:10.3390/e25121605_

Round 1

Reviewer 1 Report

Comments and Suggestions for Authors

If I understand the paper correctly, then the author proposes a measure of how easy or difficult it is for a system to escape from a region within its phase space.

Under hard ergodicity breaking, where different regions of phase space are mutually inaccessible, in order to know where a system will be in the long term, all we need to know is one state, and we have our answer with certainty. An example would be a roll of a die, where the state of the system is forever fixed at the points rolled.

But systems with weaker forms of ergodicity breaking are different. For instance, a finite-size Ising model below the critical temperature will be trapped for long times in either a predominantly positive or negative-magnetization state. However, because of the finite size of the system, it is possible for the macroscopic magnetization to flip. In the very long run, the system does visit all possible states but it will be trapped for long periods in one or the other basin of attraction.

The author's measure is for situations akin to this second case, where we may wish to quantify how likely it is (or how long we may have to wait) to escape from a given basin of attraction.

I think this is a nice question to ask and discuss, and I recommend publishing the paper.

Reviewer 2 Report

Comments and Suggestions for Authors

This work proposes a measure “cohesion” to represent the amount of epistemic uncertainty that can be reduced. In general, this article is well-written and the demonstration on the case study is clear. I believe this work can have scientific contribution to the sensitivity analysis community. Therefore, I suggest that this article can be published as it is.

Comments on the Quality of English Language

Good.